# Clinical Behavior of Short Dental Implants: Systematic Review and Meta-Analysis

**DOI:** 10.3390/jcm9103271

**Published:** 2020-10-12

**Authors:** Andrea Torres-Alemany, Lucía Fernández-Estevan, Rubén Agustín-Panadero, José María Montiel-Company, Carlos Labaig-Rueda, José Félix Mañes-Ferrer

**Affiliations:** Department of Dental Medicine, Faculty of Medicine and Dentistry, University of Valencia, C/Gascó Oliag 1, 46010 Valencia, Spain; anto.aleny@gmail.com (A.T.-A.); ruben.agustin@uv.es (R.A.-P.); jose.maria.montiel@uv.es (J.M.M.-C.); Carlos.Labaig@uv.es (C.L.-R.); Jose.Manes@uv.es (J.F.M.-F.)

**Keywords:** short implants, crown-to-implant ratio, survival rate, edentulous mouth, partially edentulous mouth

## Abstract

Short implants are an increasingly common alternative to other surgical techniques in areas where bone availability is reduced. Despite the advantages they offer, a variety of biological repercussions have been described in the literature that can even lead to the loss of these. The aim of this systematic review and meta-analysis was to analyze the impact of the use of short implants on their survival and on peri-implant bone loss, evaluating the influence that length, diameter, and crown-to-implant ratio (C/I) have on these parameters. This systematic review was based on guidelines proposed by the Preferred Reporting Items for Systematic Reviews and Meta-Analyses (PRISMA). An electronic search was conducted using terms related to the use of short implants in partially or totally edentulous patients. A total of five databases were consulted in the literature search: PubMed, Embase, Cochrane, Scopus, and Web of Science. After eliminating the duplicate articles and assessing which ones met the inclusion criteria, 15 articles were included for the qualitative analysis and 14 for the quantitative study. Through meta-analysis, the percentage of implant loss and peri-implant bone loss was estimated. Relating these parameters to length, diameter, and C/I ratio, no significant differences have been found for implant loss (values of *p* = 0.06, 0.10, and 0.9, respectively for length, diameter, and C/I ratio), nor for peri-implant bone loss (values of *p* = 0.71, 0.72, and 0.36, respectively for length, diameter, and C/I ratio). In conclusion, the use of short implants does not seem to have a significant influence on marginal bone loss or the survival rate of implants.

## 1. Introduction

The key to the success of dental implants is due to their osseointegration capacity [1], which can be defined as the direct contact of the implant surface with the bone, without the interposition of fibrous tissue. For good osseointegration, the implants must remain immobile when loading the prosthesis, and there may be vertical bone loss of up to 0.2 mm during the first year. When this bone loss is greater, there is a problem of osseointegration and periimplantitis may occur [2].

On the other hand, the teeth during their function perform a series of tension and compression forces that are transmitted to the surrounding alveolar bone. These forces cause the bone to be continuously stimulated, which is necessary to maintain its shape and density.

Therefore, when there is an absence of teeth, this lack of stimulation causes a decrease in bone density and volume, leading to a progressive reabsorption of the alveolar bone which in time leads to atrophy of the jaws [3]. This loss of bone volume at the level of the posterior maxilla and mandible can complicate rehabilitation with implants, since it leads to a reduction in the distance to the maxillary sinus and the lower dental nerve, respectively [4].

Various techniques have been used for many years to avoid damaging these structures, such as alveolar ridge augmentation procedures, bone grafting, tooth nerve transposition, or even zygomatic implants [1,4,5,6]. Bone grafting is one of the most widely used procedures, with implant success rates of up to 89% [7]. Although these procedures have high success rates, they can also have some drawbacks such as post-operative infections, collapse of mucosal tissue, pain, bleeding, and neurosensory deficits [8]. These are also complex procedures and advanced surgical techniques not applicable to all patients or by all professionals.

An alternative to these surgical treatments is the use of short implants in areas where bone availability is reduced. This treatment option has been included in numerous studies, using the survival of these implants as the main study variable [1,8,9,10,11].

The definition of a short implant has been a widely debated issue throughout the literature. Initially, short implants were considered to be those with a length of <11 mm. Later it has evolved and there are authors who consider them short when their length is <7 mm and others when it is <8 mm [6,12,13]. In 2016, the European Consensus Conference established that short implants were those with an intraosseous length ≤8 mm and a diameter ≥3.75 mm [14].

Short implants have a number of advantages over standard implants; with short implants, less vertical bone grafting is performed, which means less time and cost of treatment and less morbidity for the patient. In addition, there is less surgical risk of perforating the maxillary sinus, of producing paresthesias due to dental nerve injury, of causing damage due to overheating during osteotomy, and of injuring the root of adjacent teeth.

Some studies suggest that the survival rate of short implants in the original bone is the same as that of standard implants placed through bone regeneration procedures [15,16]. But it should be noted that historically, short implants have been associated with a lower survival rate, with unpredictable results [6].

These implants also tend to have their crowns increased in length in order to establish occlusion with their antagonist, which causes an increase in the crown/implant ratio (C/I) [12,17].

Specifically, with short implants, increasing the height of the crown creates an unfavorable C/I ratio since the lever arm increases and when non-axial forces are received there is an increase in tension in the basal bone [12,17]. This is why the C/I ratio was originally considered to be 1:1 as in natural teeth. However, some authors now suggest the possibility of using implants with a C/I ratio greater than 1:1 without presenting long-term complications [18,19,20,21].

Based on the background described, the use of short implants, like all implants, can generate a series of complications in the patient among which we highlight, due to their relevance, the loss of the implant, and the loss of marginal bone. The use of short implants can be conditioned, among other things, by three aspects: the length of the implant itself, its diameter, and the C/I ratio generated with respect to the prosthesis it supports. Therefore, the main objective of this systematic review has been to analyze the available scientific evidence about the impact of the use of short implants in terms of implant survival and peri-implant bone loss. As secondary objectives, we set out to see if the scientific evidence provides us with information about the influence of length, diameter, and C/I ratio of short implants in terms of the biological complications mentioned above.

## 2. Materials and Methods

This bibliographic search was conducted following Preferred Reporting Items for Systematic Reviews and Meta-Analyses (PRISMA) guidelines for systematic reviews and meta-analyses. The review was also registered in the PRISMA database (PROSPERO: international prospective register of systematic reviews), registration number: CRD42020191093.

The population, intervention, comparison, outcome (PICO) question was: “Is the use of short implants with an increased crown-to-implant ratio an adequate treatment option compared to other treatment options in partially or totally edentulous patients?” with the following components: population: partially or totally edentulous patients; intervention: short implant placement with increased crown-to-implant ratio; comparison: other options that improve that crown-to-implant ratio, such as regeneration techniques or overdentures; and outcomes: implant failure/success, implant survival rate, prosthetic or biological complications.

An electronic search was conducted in the following databases: PubMed, Embase, Cochrane, Scopus and Web of Science. In order to focus the results on our PICO question, the search included several MeSH (Medical Subject Headings) terms: “mouth”, “edentulous”, “jaw”, “bone regeneration”, “bone transplantation”, “denture”, and “overlay.” The Boolean operators applied were “OR” and “AND.” The search terms were structured as follows: ((mouth, edentulous) OR (jaw, edentulous)) AND ((bone regeneration) OR (bone transplantation)) OR ((denture, overlay)).

Two researchers (A.T.A.; J.M.F.) conducted the database searches in duplicate independently. Titles and abstracts were selected by applying inclusion and exclusion criteria. One researcher (A.T.A) extracted data for relevant variables. The systematic review was carried out by (A.T.A) and the posterior meta-analysis was performed by a researcher not involved in the selection process (J.M.C).

Inclusion criteria: partially or totally edentulous patients treated with implants at the posterior level both maxillary and mandibular, studies with a minimum sample size of 10 patients, studies with a follow-up time of at least one year, studies using implants with a length of less than or equal to 8 mm as opposed to implants with a length greater than 8 mm; in terms of studio design: randomized clinical trials (RCTs) and prospective case-control studies (PCC) and studies analyzing implant survival rate, implant failure, bone loss, presence of periimplantitis.

Exclusion criteria: animal or in vitro studies, pre-clinical studies and those describing clinical cases, case series, prospective cohort studies and retrospective studies, studies with insufficient information, those that use the same population group (since in this case we will only use the study with the longest follow-up period), and the studies in which the prosthesis is supported at the same time by short and standard implants.

The variables registered in each of the studies were: author and year of publication; type of study; years of follow-up; number of patients and initial implants; location, length, and diameter of the implants; type of restoration (cemented or screwed); splinted or non-splinted restorations; prosthetic complications; mean marginal bone loss and standard deviation (SD); number of lost implants; percentage of implant survival; and crown-to-implant ratio (C/I ratio).

The quality of the studies was independently analyzed by the same researchers. To evaluate the quality of the cohort studies the Newcastle–Ottawa Quality Evaluation Scale (NOS) was used [22].

The biological repercussions studied for the meta-analysis were the percentage of implant loss and the mean marginal bone loss around the implant. The studies were combined using a random effects model with the inverse of the variance method. For all the estimated variables we calculated their 95% confidence interval and their prediction interval. Heterogeneity among the combined studies was assessed by Q test (*p* < 0.05) and was quantified with I2 considering a slight heterogeneity if it was between 25–50%, moderate between 50–75%, and high if >75%. A subgroup analysis was performed to assess the existence of differences due to diameter, length, C/I ratio, and follow-up time using the Q test; and a meta-regression was performed using the mixed effects model. The existence of significant variables was also assessed by means of the QM value (Q test of moderators value). The existence of statistical significance was established for a *p* < 0.05. The meta-analyses have been represented with forest plot; and the publication bias has been assessed by means of the Trim and Fill adjustment method, and are represented with Funnel plots.

## 3. Results

The initial electronic search identified 300 articles in PubMed, 126 in Embase, 110 in Cochrane, 120 in Scopus, and 65 in Web of Science; which together with the 20 articles derived from the manual search make a total of 741 articles. Of all the articles derived from the initial search, we proceeded to manually eliminate the articles that were duplicated in the different databases. In this way, a total of 89 articles were eliminated. Once we eliminated the duplicates, we proceeded to read the title and abstract of the remaining 652 articles, of which 513 were excluded. After this exclusion, we were left with 139 potential articles to include in this review for full text reading. After reading these 139 articles, we proceeded to discard 124 articles for not presenting relevant data for the review and for not meeting the inclusion criteria described above; that is why the review finally included a total of 15 articles, which meet the inclusion criteria, and from which the qualitative analysis has been carried out. In order to carry out the quantitative analysis, we discarded one of the articles because of insufficient information, leaving 14 articles. In Figure 1 is the PRISMA flow chart, where the selection process of the articles is represented.

The results of methodological quality assessment using the Newcastle–Ottawa (NOS) Scale are shown in Table 1. We can see that the quality of the selected studies is high, being all of them above the value 7.

Criteria: (1) Representativeness of the exposed cohort. (2) Selection of the non-exposed cohort. (3) Ascertainment of exposure. (4) Demonstration that outcome of interest was not present at start of study. (5) Comparability of cohorts on the basis of the design or analysis controlled for confounders: for the most important factor (5a), for other factors (5b). (6) Assessment of outcome. (7) Adequate follow-up time. (8) Adequacy of follow-up of cohorts.

The quantitative synthesis included 14 articles (Table 2). The variables presented in the table are: author and year of publication, type of study, years of follow-up, number of initial implants, length and diameter of the implants, mean marginal bone loss and standard deviation (SD), number of lost implants, crown-to-implant ratio (C/I ratio) and the assessment of the articles according to the Newcastle–Ottawa Scale.

In the table we indicate the type of article that has been included, whether they are prospective control case studies (PCC) or randomized clinical trials (RCT). Through quantitative analysis we wanted to estimate on the one hand the percentage of implant loss, and on the other hand, the average marginal bone loss around the implant.

Both parameters were related to the length and diameter of the implant, as well as the C/I ratio, which are the fundamental objectives of our work.

To carry out this analysis we eliminated an article, Kim et al. 2015 [23], because it presented insufficient information to obtain these data.

These two variables were studied separately, using 14 studies to determine the percentage of implant failure or loss, and 10 studies to estimate the mean marginal bone loss.

### 3.1. Percentage of Failure or Implant Loss

Data from 14 studies have been combined using a random effects model with the inverse of variance method (Figure 2), estimating an implant failure rate of 1.3% (Confidence Interval-95% between 0.63% and 1.97%) and a prediction interval between 0.61% and 2%. No heterogeneity was detected among the combined studies (test Q = 22.7; *p*-value = 0.909; I^2^ = 0%). 

### 3.2. Analysis by Subgroups According to C/I Ratio

The percentage of failure in implants with a C/I ratio = 1 was 1.03%, with C/I ratio = 2 it was 1.36%, and finally for a C/I ratio = 3 it was 0.87% (Figure 3).

There were no significant differences between the subgroups (Q test = 21.3; *p*-value = 0.902).

### 3.3. Analysis by Subgroups According to Diameter

The percentage of failure in implants with standard diameter was 1.28%, while with narrow diameter it was 12.9% (Figure 4), although no significant differences between subgroups were detected (Q test = 2.67; *p*-value = 0.102).

### 3.4. Analysis by Subgroups According to Length

The failure rate for short implants was 2.11%, while with long implants it was 0.8% (Figure 5). There were no significant differences between the subgroups (Q test = 3.46; *p*-value = 0.063) although its value was very close to significance.

### 3.5. Meta-Regression

Using a combination model of mixed effect studies, the C/I ratio, diameter, implant length, and years of follow-up have been analyzed in the proportion of implant loss (Table 3). The moderator’s test showed QM value = 14.38 and *p*-value = 0.230, indicating that none of the variables were significant in the model.

### 3.6. Publication Bias

The Trim and Fill method was used to assess the publication bias (Figure 6). Using a random effects model (inverse variance method), five studies were added to adjust the asymmetry of the Funnel plot, obtaining an estimate of the failure rate of 1.18% (CI-95% between 0.52% and 1.87%) which does not differ from the initial estimate of 1.3% (CI-95% between 0.63% and 1.97%), making it possible to rule out the existence of publication bias.

### 3.7. Mean Marginal Bone Loss from the Implant

Data from 10 studies were combined using a random effects model with the inverse method of variance, estimating a mean marginal bone loss to the implant of 0.58 mm (95% CI between 0.40 mm and 0.75 mm) and a predictive range of -0.36 mm to 1.52 mm (Figure 7). There is high heterogeneity among the combined studies (test Q = 965.9; *p*-value < 0.01; I^2^ = 97%).

### 3.8. Analysis by Subgroups According to C/I ratio

The mean marginal bone loss to the implant when C/I ratio = 1 was 0.84 mm, with C/I ratio = 2 was 0.54 mm, and finally for a C/I ratio = 3 of 0.44 (Figure 8). No significant differences were detected between the three subgroups (Q test = 2.05; *p*-value = 0.359).

### 3.9. Analysis by Subgroups According to Diameter

The average marginal bone loss to the implant with standard diameter was 0.52 mm, while with narrow diameter it was 0.58 mm (Figure 9). Although no significant differences between the subgroups were detected (Q test = 0.13; *p*-value = 0.715).

### 3.10. Analysis by Subgroups According to Length

The average marginal bone loss to the implant of short length was 0.54 mm, while with long length it was 0.61 mm (Figure 10). No significant differences between the long and short implant subgroups were determined (Q test = 0.14; *p*-value = 0.713).

### 3.11. Meta-Regression

Using a combination model of mixed effect studies, the C/I ratio, diameter, implant length, and years of follow-up on mean marginal bone loss to the implant were analyzed (Table 4). The moderator test showed a QM value of 1.82 and *p* = 0.872, indicating that none of the variables showed significance in the model.

### 3.12. Publication Bias

The Trim and Fill method to adjust the asymmetry of the Funnel plot was used adding 11 new studies to those initially combined to obtain a symmetrical image. The new estimate of the mean marginal bone loss to the implant was 0.32 mm (IC-95% between 0.12 and 0.51 mm) which differs slightly from the mean initially obtained and estimated at 0.58 mm (IC-95% between 0.40 mm and 0.75 mm). The Figure 11 shows the two Funnel plots (initial and adjusted) indicating a slight but not significant difference between their confidence intervals and indicating a small probability of publication bias.

## 4. Discussion

Nowadays, it is very common to use short implants in order to avoid surgical techniques that may represent a risk for important anatomical structures. As we have pointed out, in the literature there have been many diverse opinions about the complications that may result from the use of short implants when there is an increased crown-to-implant (C/I) ratio, such as the marginal bone loss that may occur, or even the loss of the implants.

According to the results obtained in the quantitative analysis, it has been estimated that the percentage of implant loss is 1.3 % and the average annual marginal bone loss is 0.58 mm. It has been established that none of the variables to which both parameters were related, such as length, diameter, C/I ratio, and follow-up time, affect the results significantly.

Adánez [24] and Guarnieri [25] in their studies intended to compare the marginal bone loss between short and standard implants. They also evaluated the survival of short implants in relation to the C/I ratio. Finally, both found that there was no significant difference between implant survival rate and marginal bone loss when the C/I ratio was increased. The same conclusion was reached by Sarhmann [26] and Hadzik [27], whose studies determined that the C/I ratio had no significant influence on the implant survival rate and marginal bone loss.

Ghariani et al. [28] also wanted to evaluate the survival rate and marginal bone loss on implants with different C/I ratios. In this case, although there was a trend towards greater marginal bone loss with an increased C/I ratio, the results were not statistically significant. Nor did they see any influence on implant survival.

In 2017, Malchiodi et al. [29] conducted a study in which they analyzed the survival rate and marginal bone loss between short and standard implants. They found that the C/I ratio was not significantly related to bone loss or implant loss. Neither the diameter nor the length of the implants had any influence.

Mangano et al. [30] aimed to evaluate the influence that C/I ratio had on marginal bone loss, on prosthetic complications, and on the survival of short implants. Although complications were more frequent in restorations with a C/I ratio greater than 2, no significant differences were seen. Nor was there any difference in marginal bone loss or survival rate.

Pohl et al. [31] studied the survival of short implants compared to standard implants over a 3-year follow-up period. They observed no relationship between implant survival rate and implant length. Nor was there any relationship with marginal bone loss.

Schincaglia et al. [32] compared short implants versus standard implants undergoing a sinus lift procedure. In this case, although the C/I ratio was found to be statistically significant between both groups, this had no effect on marginal bone loss. There was also no significant correlation between the survival rate of implants with respect to the length of both groups.

On the other hand, some authors found significant differences between the parameters studied, and the use of short implants which may have a greater impact.

Malchiodi et al. [33] in 2014 conducted a study in which they wanted to determine the influence of the C/I ratio on bone loss and implant survival rate. To do this they used more than 150 short implants, comparing them with standard implants. In this case, they saw a significant relationship between marginal bone loss and implant loss when the C/I ratio was increased. These results are contrary to those studied by the same authors in 2017 [29]. This could be mainly due to the difference in the number of implants in both studies, as in 2014 there were 259 short and standard implants, while in 2017 there were 113 implants. The follow-up time for both studies was three years.

In the study by Naenni et al. [34] they aimed to assess whether short implants had similar survival and bone loss rates as standard implants. In this case, significant differences were seen between implant length and survival rate, with short implants having a lower survival rate than standard implants. Significant differences were also seen in the C/I ratio between the two groups, although this had no influence on marginal bone loss. These differences in implant loss, according to the authors, can be attributed to the surfaces of the implants and the loading protocols used, which may vary from one study to another.

Rossi et al. [35] wanted to compare the clinical and radiographic findings obtained between short and standard implants over five years of follow-up. This study showed similar marginal bone loss in both groups, however, implant loss was seen to be greater in the short group. The authors attribute this event to the fact that three of the four short implants that failed were placed in the maxilla, and according to them, survival is slightly higher in the mandible compared to the maxilla.

Finally, we also found studies that show quite positive results regarding the use of short implants. In the case of Zadeh et al. [36] their main objective was to study whether bone loss occurred in a similar way in both groups of implants. They concluded that marginal bone loss was significantly lower in short implants than in standard implants. This, they said in the study, could have occurred as a consequence of overheating at the time of insertion of the long implants.

Blanes et al. [37] wanted to evaluate the influence of C/I ratio on marginal bone loss and the long-term survival rate of implants. In this case, they established an inverse relationship, it was seen that those restorations with an increased C/I ratio had less bone loss than those with a decreased C/I ratio. This could be attributed to the follow-up time, which in this study was 10 years.

As we can see, most of the studies analyzed conclude that there is no relationship between the use of short implants with an increased C/I ratio and marginal bone loss or implant survival. These results coincide with the quantitative analysis of our study.

However, when interpreting the results of the qualitative analysis, we must consider certain limitations. Among them is the time needed to follow up on the studies. Although most of them were between three and five years [25,26,27,29,30,31,33,34,35,36], there are several that also have a follow-up of only one year [24,28,32]. Given the absence of long-term longitudinal studies of more than 10 years on short implants it is difficult to define whether the passage of time affects the survival rate of short implants.

Finally, the splinting of the restorations should also be considered. It has been seen that when they are splinted there is a better distribution of the occlusal loads between the implants, thus reducing stress and increasing the stability of the restorations [37,38]. In some of the studies analyzed, the authors use both simple and splinted crowns [25,29,30,33,37], but only one of them specifies the number of short and standard implants that use one or the other [25]. Therefore, it is not possible to analyze the influence of the C/I ratio with this parameter.

## 5. Conclusions

After having systematically reviewed the scientific literature regarding the use of short implants for the prosthetic rehabilitation of partially or totally edentulous patients, we can conclude, taking into account the limitations of this study, the following:-Short implants have a survival and bone loss rate similar to those of conventional length.-As for the parameter implant length, no statistical significance has been established in terms of its influence on bone or implant loss, but in the latter case (implant loss) a value close to statistical significance is evident.-The parameter implant diameter has not been established as statistically significant in terms of its influence on bone or implant loss.-Finally, the parameter crown/implant ratio of the implants has also not been established as being statistically significant in terms of its influence on bone or implant loss.

## Figures and Tables

**Figure 1 jcm-09-03271-f001:**
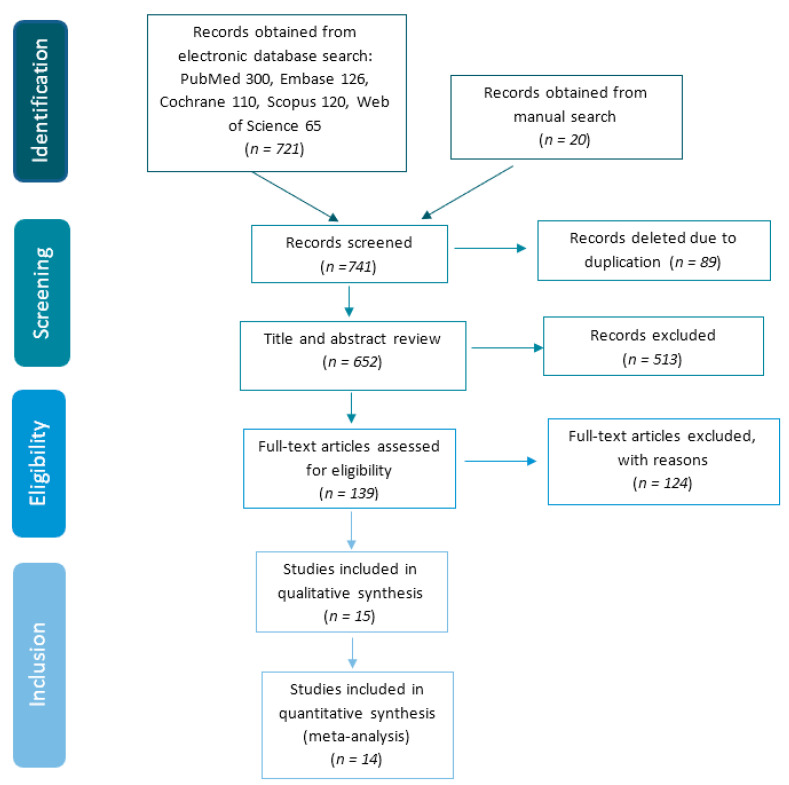
Flow chart of study selection procedure.

**Figure 2 jcm-09-03271-f002:**
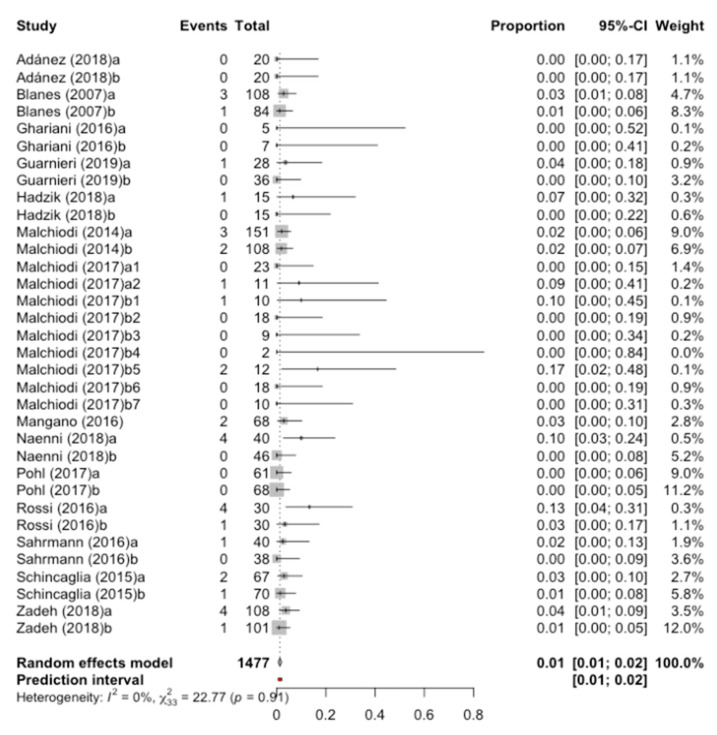
Forest plot of the meta-analysis of the implant failure rate. CI is the 95% confidence interval of the percentage of failure or implant loss.

**Figure 3 jcm-09-03271-f003:**
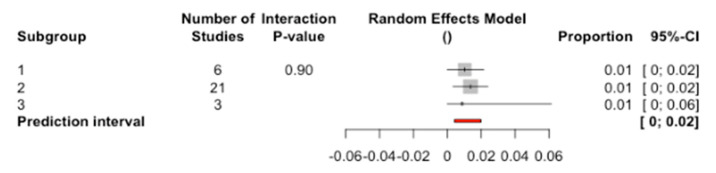
Forest plot of the meta-analysis of the percentage of implant failure according to crown-to-implant (C/I) ratio.

**Figure 4 jcm-09-03271-f004:**
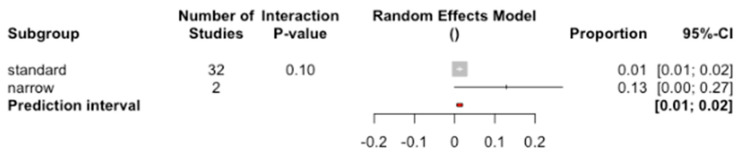
Forest plot of the meta-analysis of the percentage of implant failure according to diameter.

**Figure 5 jcm-09-03271-f005:**
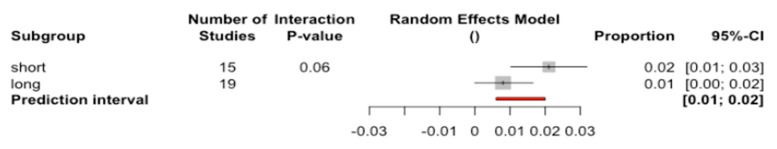
Forest plot of the meta-analysis of the percentage of implant failure according to the length.

**Figure 6 jcm-09-03271-f006:**
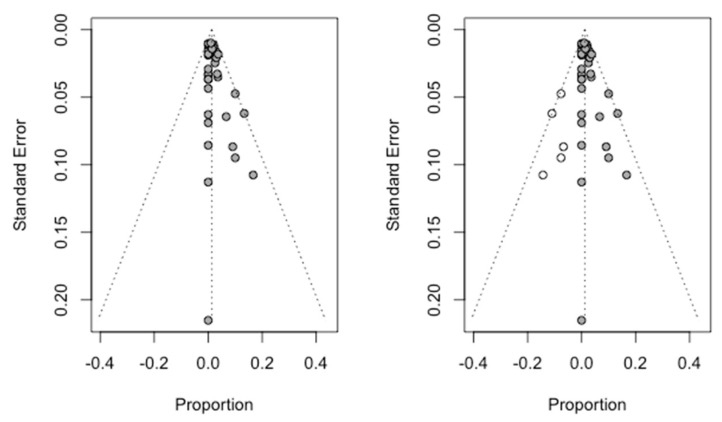
Initial funnel plot and after Trim and Fill adjustment of the implant failure percentage.

**Figure 7 jcm-09-03271-f007:**
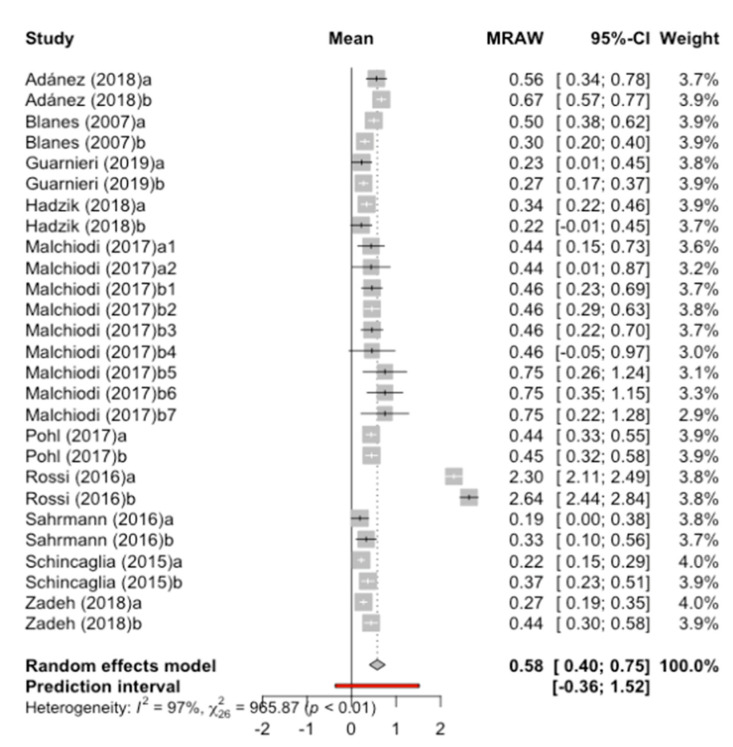
Forest plot of the meta-analysis of marginal bone loss at the implant. MRAW is the crude mean.

**Figure 8 jcm-09-03271-f008:**
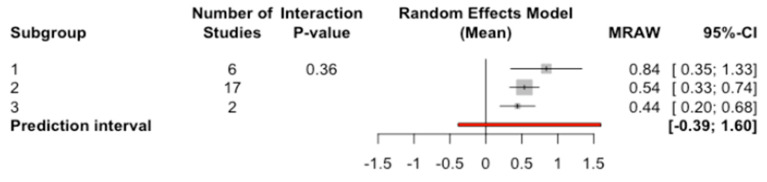
Forest plot of the meta-analysis of marginal bone loss at the implant according to C/I ratio.

**Figure 9 jcm-09-03271-f009:**
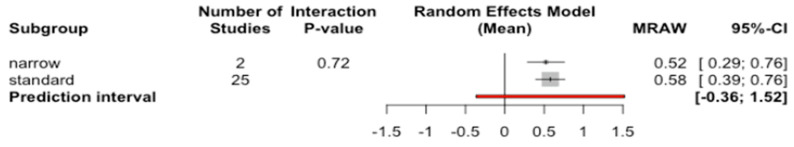
Forest plot of the meta-analysis of marginal bone loss to the implant according to diameter.

**Figure 10 jcm-09-03271-f010:**
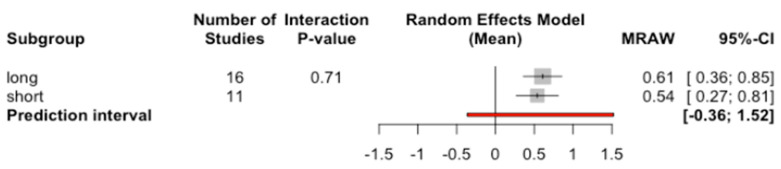
Forest plot of the meta-analysis of marginal bone loss to the implant according to length.

**Figure 11 jcm-09-03271-f011:**
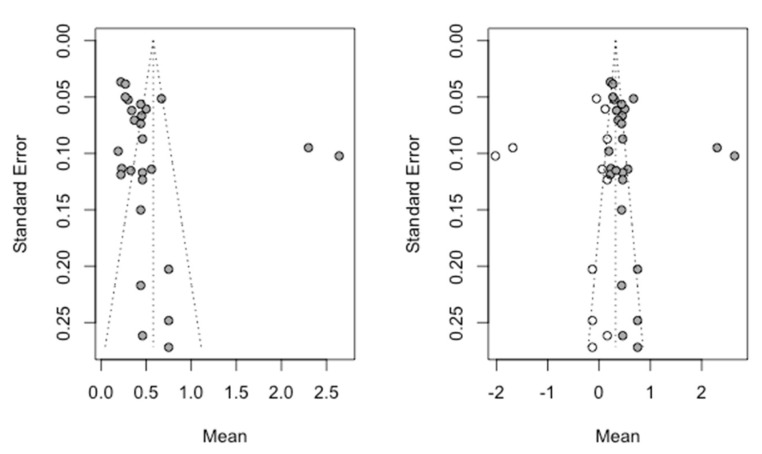
Funnel plot initial and after Trim and Fill adjustment.

**Table 1 jcm-09-03271-t001:** Quality of the studies in the Newcastle–Ottawa Scale for cohort studies.

AUTHOR (Year)	SELECTION (****)				COMPARABILITY (**)		OUTCOMES (***)			TOTAL
	1	2	3	4	5a	5b	6	7 (1 year)	8	
Adánez et al. (2018)		*	*	*	*	*	*	*	*	8
Blanes et al. (2007)	*	*	*	*	*	*	*	*		8
Ghariani et al. (2016)		*	*	*	*		*	*	*	7
Guarnieri et al. (2019)	*	*	*	*	*	*	*	*	*	9
Hadzik et al. (2018)	*	*	*	*	*		*	*		7
Malchiodi et al. (2014)	*	*	*	*	*	*	*	*	*	9
Malchiodi et al. (2017)	*	*	*	*	*		*	*	*	8
Mangano et al. (2016)	*	*	*	*	*		*	*	*	8
Naenni et al. (2018)	*	*	*	*	*	*	*	*	*	9
Pohl et al. (2017)	*	*	*	*	*	*	*	*		8
Rossi et al. (2016)	*	*	*	*	*	*	*	*	*	9
Sahrmann et al. (2016)	*	*	*	*	*	*	*	*	*	9
Schincaglia et al. (2015)	*	*	*	*	*	*	*	*	*	9
Zadeh et al. (2017)	*	*	*	*	*	*	*	*	*	9

* This symbol corresponds to the star given in each of the 8 items represented on the NOS scale. In the case of the selection category, up to 4 stars can be given (****), in the comparability category a maximum of 2 stars can be given (**); and in the case of the results up to 3 stars will be given (***). The more stars awarded in total, the higher the quality of the articles.

**Table 2 jcm-09-03271-t002:** Quantitative analysis of articles included in the systematic review.

AUTHOR (YEAR)	TYPE OF STUDY	YEARS OF FOLLOW-UP	LENGTH * (mm)	DIAMETER ** (mm)	No. INITIAL IMPLANTS	MEAN MARGINAL BONE LOSS (mm)	SD	S/NS RESTORATIONS	No. LOST IMPLANTS	C/I RATIO	NOS
**Adánez (2018)**	PCC	1	Short	Standard	20	0′56	0′51	S	0	2	8
			Standard	Standard	20	0′67	0′23	S	0	1	
**Blanes (2007)**	PCC	10	Short	Standard	108	0′5	0′63	S and NS	3	1	8
			Standard	Standard	84	0′3	0′48	S and NS	1	2	
**Ghariani (2016)**	PCC	1	Short	Standard	5			NS	0	3	7
			Standard	Standard	7			NS	0	2	
**Guarnieri (2019)**	PCC	3	Short	Standard	28	0′23	0′6	S and NS	1		9
			Standard	Standard	36	0′27	0′3	S and NS	0		
**Hadzik (2018)**	PCC	3	Short	Standard	15	0′34	0′24	NS	1	2	7
			Standard	Standard	15	0′22	0′46	NS	0	2	
**Malchiodi (2014)**	PCC	3	Short	Standard	151			S and NS	3		9
			Standard	Standard	108			S and NS	2		
**Malchiodi (2017)**	PCC	3	Short	Standard	23	0′44	0′72	S and NS	0	3	8
			Short	Standard	11	0′44	0′72	S and NS	1	3	
			Standard	Narrow	10	0′46	0′37	S and NS	1	2	
			Standard	Standard	18	0′46	0′37	S and NS	0	2	
			Standard	Standard	9	0′46	0′37	S and NS	0	2	
			Standard	Standard	2	0′46	0′37	S and NS	0	2	
			Standard	Narrow	12	0′75	0′86	S and NS	2	2	
			Standard	Standard	18	0′75	0′86	S and NS	0	2	
			Standard	Standard	10	0′75	0′86	S and NS	0	2	
**Mangano (2016)**	PCC	5	Short	Standard	68			S and NS	2	2	8
**Naenni (2018)**	RCT	5	Short	Standard	40			NS	4	2	9
			Standard	Standard	46			NS	0	2	
**Pohl (2017)**	RCT	3	Short	Standard	61	0′44	0′44	NS	0	2	8
			Standard	Standard	68	0′45	0′55	NS	0	1	
**Rossi (2016)**	PCC	5	Short	Standard	30	2′3	0′52	NS		2	9
			Standard	Standard	30	2′64	0′56	NS		1	
**Sahrmann (2016)**	RCT	3	Short		40	0′19	0′62	NS	1	2	9
			Standard		38	0′33	0′71	NS	0	2	
**Schincaglia (2015)**	RCT	1	Short	Standard	67	0′22	0′3	NS	2	2	9
			Standard	Standard	70	0′37	0′59	NS	1	1	
**Zadeh (2018)**	RCT	3	Short	Standard	108	0′27	0′4	S	4	2	9
			Standard	Standard	101	0′44	0′74	S	1	1	

SD: standard deviation; C/I ratio: crown-to-implant ratio; S: splinted; NS: non-splinted; NOS: Newcastle–Ottawa Scale; PCC: prospective control case studies; RCT: randomized clinical trial. * Length: short ≤ 8 mm and standard > 8 mm. ** Diameter: standard (≥3.75 mm) and narrow < 3.75 mm.

**Table 3 jcm-09-03271-t003:** Estimation of the moderators of the meta-regression.

	Estimator	IC-95%	*p*-Value
Intercept	0.131	−0.009; 0.272	0.067
Standard diameter	−0.126	−0.267; 0.014	0.077
C/I ratio = 2	−0.005	−0.022; 0.012	0.558
C/I ratio = 3	−0.016	−0.072; 0.040	0.577
Years of follow-up	0.001	−0.002; 0.004	0.485
Short length	0.017	−0.001; 0.035	0.055

CI is the 95% confidence interval of the percentage of implant loss according to diameter, C/I ratio and length of the implants.

**Table 4 jcm-09-03271-t004:** Estimation of the moderators of the meta-regression.

	Estimator	IC-95%	*p*-Value
intercept	0.975	−0.002; 1.952	0.050
Standard diameter	−0.084	−0.937; 0.768	0.845
C/I ratio = 2	−0.322	−0.849; 0.204	0.230
C/I ratio = 3	−0.453	−1.424; 0.517	0.360
Years of follow-up	−0.019	−0.114; 0.075	0.685
Short length	0.061	−0.433; 0.556	0.807

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
