# Peer review of "Clinical Behavior of Short Dental Implants: Systematic Review and Meta-Analysis"

_jcm, 2020, doi:10.3390/jcm9103271_

Round 1

Reviewer 1 Report

What makes this paper different from other systemic reviews in the past includes papers comparing bone loss and survival rates of short and standard implants over long period.
It would be nice if the following points were additionally explained.
The survival rate of short implants varies depending on the follow-up time, and it seems necessary to organize this topic more systematically (rather than listing the outcome of the papers) and explain the reason.
In the study of Rossi et al [35], there is no difference in maringal bone loss, but the survival rate of short implant showed difference. If there is an additional explanation for this outcome, it would be good for readers to understand.

Author Response

Response to Reviewer 1

Comments and Suggestions for Authors

“What makes this paper different from other systemic reviews in the past includes papers comparing bone loss and survival rates of short and standard implants over long period.
It would be nice if the following points were additionally explained.
The survival rate of short implants varies depending on the follow-up time, and it seems necessary to organize this topic more systematically (rather than listing the outcome of the papers) and explain the reason.”

Dear Reviewer, Thank you for your comment. The text has been revised in order to explain this subject in a more systematic way. The following information has been added to the text.

“…As we can see, most of the studies analyzed conclude that there is no relationship between the use of short implants with an increased C/I ratio and marginal bone loss or implant survival. These results coincide with the quantitative analysis of our study.

However, when interpreting the results of the qualitative analysis, we must consider certain limitations. Among them is the time needed to follow up on the studies. Although most of them are between 3 and 5 years old (25-27,29-31,33-36), there are several that also have a follow-up of only 1 year (24,28,32). Given the absence of long-term longitudinal studies of more than 10 years on short implants it is difficult to define whether the passage of time affects the survival rate of short implants..”

“In the study of Rossi et al [35], there is no difference in maringal bone loss, but the survival rate of short implant showed difference. If there is an additional explanation for this outcome, it would be good for readers to understand”.

Dear Reviewer, Thank you for your comment. The studies have been reviewed introducing the following information as you advice.

“…Rossi et al [35] wanted to compare the clinical and radiographic findings obtained between short and standard implants over 5 years of follow-up. This study showed similar marginal bone loss in both groups, however, implant loss was seen to be greater in the short group. This could also be attributed to the follow-up time of the study. The authors attribute this event to the fact that 3 of the 4 short implants that failed were placed in the maxilla, and according to them, survival is slightly higher in the mandible compared to the maxilla.”

Following your advice, we have also added more information on line 181 to make it better understood.

“…In the study by Naenni et al [34] they aimed to assess whether short implants had similar survival and bone loss rates as standard implants. In this case, significant differences were seen between implant length and survival rate, with short implants having a lower survival rate than standard implants. Significant differences were also seen in the C/I ratio between the two groups, although this had no influence on marginal bone loss. The main difference between these studies and the previous ones could be the follow-up time, since in the above-mentioned studies almost all of them last between 1 and 3 years, whereas this one is 5 years. These differences in implant loss, according to the authors, can be attributed to the surfaces of the implants and the loading protocols used, which may vary from one study to another.”

Reviewer 2 Report

Personally, I have been dealing with short dental implants for almost 10 years. I have published several publications on this topic and am currently preparing the longest follow-up report on the use of short dental implants.

The article presented to me for evaluation is substantively interesting, it deals with a clinically and scientifically important subject.

The team of authors did an in-depth and acutal review of the literature correctly.

In my opinion, it will be a valuable source of knowledge for both clinicians and researchers dealing with the subject of short implants.

However,  the prosthetic superstructures on short implants are blocked - it seems worth discussing also the issue of single and splinted crowns since different are the biomechanics of single implants with a single crown and bridges or splinted restorations.

Since it is a common practice to splint crowns on short implants, the Authors should point if the crowns were single o splinted in all 14 evaluated articles. Such information should be added in the table and explained in the discussion.  Splinting the crowns when examining the impact of C/I ratio would change the distribution of forces and consequently, the results would be disturbed. 

Author Response

Response to Reviewer 2

Comments and Suggestions for Authors

“Personally, I have been dealing with short dental implants for almost 10 years. I have published several publications on this topic and am currently preparing the longest follow-up report on the use of short dental implants.
The article presented to me for evaluation is substantively interesting, it deals with a clinically and scientifically important subject.
The team of authors did an in-depth and acutal review of the literature correctly.
In my opinion, it will be a valuable source of knowledge for both clinicians and researchers dealing with the subject of short implants.

However, the prosthetic superstructures on short implants are blocked - it seems worth discussing also the issue of single and splinted crowns since different are the biomechanics of single implants with a single crown and bridges or splinted restorations.

Since it is a common practice to splint crowns on short implants, the Authors should point if the crowns were single o splinted in all 14 evaluated articles. Such information should be added in the table and explained in the discussion.  Splinting the crowns when examining the impact of C/I ratio would change the distribution of forces and consequently, the results would be disturbed.”

Dear Reviewer, Thank you for your comment. Following your advice we have specified in the table those studies that use single or splinted crowns. In addition, we have added more information about splinted restorations in line 207.

“…Finally, the splinting of the restorations should also be considered. It has been seen that when they are splinted there is a better distribution of the occlusal loads between the implants, thus reducing stress and increasing the stability of the restorations (37,38). In some of the studies analyzed, the authors use both simple and splinted crowns (25,29,30,33,37), but only one of them specifies the number of short and standard implants that use one or the other (25). Therefore, it is not possible to analyze the influence of the C/I ratio with this parameter.”